# Potential Effect of Cinnamaldehyde on Insulin Resistance Is Mediated by Glucose and Lipid Homeostasis

**DOI:** 10.3390/nu17020297

**Published:** 2025-01-15

**Authors:** Marisa Jadna Silva Frederico, Paola Miranda Sulis, Landerson Lopes Pereira, Diana Rey, Marcela Aragón, Fátima Regina Mena Barreto Silva

**Affiliations:** 1Instituto de Bioeletricidade Celular (IBIOCEL): Ciência & Saúde, Departamento de Bioquímica, Centro de Ciências Biológicas, Universidade Federal de Santa Catarina, Rua João Pio Duarte Silva, 241, Sala G 301, Florianópolis 88038-000, SC, Brazil; marisafrederico@ufc.br (M.J.S.F.); pasulis@hotmail.com (P.M.S.); landersonplopes@gmail.com (L.L.P.); dprey@unal.edu.co (D.R.); 2Laboratório de Bioquímica e Farmacologia, Departamento de Farmacologia e Fisiologia, Núcleo de Pesquisa e Desenvolvimento de Medicamentos, Escola de Medicina, Universidade Federal do Ceará, Rua Coronel Nunes de Melo, Fortaleza 60430-275, CE, Brazil; 3Departamento de Farmacia, Universidad Nacional de Colombia, Av. Carrera 30 # 45-03 Edif. 450, Bogotá 111321, Colombia; dmaragonn@unal.edu.co

**Keywords:** cinnamaldehyde, insulin resistant, skeletal muscle, adipose tissue, glucose uptake, glycogen, lipids

## Abstract

Diabetes mellitus is a metabolic syndrome that has grown globally to become a significant public health challenge. Hypothesizing that the plasma membrane protein, transient receptor potential ankyrin-1, is a pivotal target in insulin resistance, we investigated the mechanism of action of cinnamaldehyde (CIN), an electrophilic TRPA1 agonist, in skeletal muscle, a primary insulin target. Specifically, we evaluated the effect of CIN on insulin resistance, hepatic glycogen accumulation and muscle and adipose tissue glucose uptake. Furthermore, the in vitro role of CIN in glucose uptake and intracellular signaling was determined in insulin-resistant rats whose calcium influx was analyzed. Moreover, the serum lipid profile was assessed following short-term CIN treatment in rats, and lipid tolerance was analyzed. The effects of CIN on insulin resistance were mediated by TRPA1, with downstream signaling involving the activation of PI3-K, MAPK, PKC, as well as extracellular calcium and calcium release from intracellular stores. Additionally, cytoskeleton integrity was required for the complete action of CIN on glucose uptake in muscle. CIN also ameliorated the serum lipid profile and improved triglyceride tolerance following acute vivo exposure.

## 1. Introduction

Diabetes mellitus (DM) is a public health disease, prompting governments to allocate yearly budgets based on an alarming estimate for 2045 [1]. Type 1 diabetes is a metabolic syndrome caused by an imbalance of glucose, lipid and proteins, due to a lack of insulin synthesis by β-cells, or the production of insufficient insulin for the regulation of intermediate metabolism. In contrast, type 2 diabetes features by insulin resistance and β-cell dysfunction, as the result of high and persistent glycemia (hyperglycemia) [2]. Although maintaining physiological levels of glycated hemoglobin (HbA1c) is important for preventing macro- and microvascular complications [3,4], a balance between glucose and plasma lipids is essential for the control of diabetes. As such, long-term therapies that can sustain glycemic control and blood lipid homeostasis, while improving insulin resistance and β-cell function, are highly sought.

Transient receptor potential ankyrin-1, a channel that is permeable to sodium (Na^+^) and calcium (Ca^2+^), has gained attention as a potential target for drug development, particularly for glucose and lipid metabolism pathways ([5]. The binding of electrophilic agonists, such as cinnamaldehyde (CIN), to TRPA1 may induce structural modifications that keep it open. Alternatively, the channel can be activated by non-electrophilic ligands through classical orthosteric ligand/receptor binding [6,7]. We previously described a role for TRPA-1 in prompt effects of CIN on insulin secretion by pancreatic islets and in its in vitro effects on glucagon-like peptide -1 release in the intestine [8]. Building on our previous studies of CIN, an electrophilic TRPA-1 agonist, we further explored its TRPA-1-dependent mechanism of action in an insulin-target tissue, namely skeletal muscle, from insulin-resistant rats. The present investigation evaluated the effects of short-term in vivo CIN administration, in rats, on insulin tolerance, hepatic glycogen accumulation, and muscle and adipose tissue glucose uptake. In addition, the in vitro intracellular signaling that transduces CIN pathways on glucose uptake was studied and calcium influx was analyzed. Furthermore, the lipid profile was assessed after short-term administration of CIN, and its acute effects were observed in vitro.

## 2. Materials and Methods

### 2.1. Rats

Adult male Wistar rats were kept with pelleted food (Nuvital, Nuvilab CR1, Curitiba, PR, Brazil) and tap water was available ad libitum. Fasting rats were deprived of food for 12 h but were handled in accordance with the ethical recommendations of the local Ethical Committee for Animal Use (Protocol CEUA/UFSC/PP00398/749).

### 2.2. Effect of Cinnamaldehyde on Insulin Resistance

To perform tests with the insulin-resistance model, rats were treated for 5 days and separated as follows: a control group, which received saline solution (0.9% sodium chloride solution, i.p. and v.o.); a Dexamethasone group, which received an intraperitoneal (s.c.) and saline (v.o.); a Dexamethasone + CIN group, which received an injection of dexamethasone (1 mg/kg/d) and CIN (20 mg/kg, v.o.); and a CIN group, which received saline (i.p.) and CIN (20 mg/kg v.o.) [9].

### 2.3. Cinnamaldehyde on Glycogen Content in Insulin-Resistant Rats

To determine glycogen, slices of liver tissue from insulin-resistant rats and rats treated with CIN (20/mg/kg) were removed at the end of the glucose tolerance curve. They were divided into four groups: Control (vehicle), dexamethasone, dexamethasone + CIN and CIN groups. Glycogen was quantified and the results were expressed as mg/dL of glycogen per mg of liver [10,11].

### 2.4. Signaling Pathway of Cinnamaldehyde on Glucose Uptake in Insulin-Resistant Rats

To evaluate the action of CIN on glucose uptake in skeletal muscle tissue and epididymal adipose tissue, tissues were collected from rats induced to insulin resistance and treated with dexamethasone and with CIN for 5 days. After the dissection and fragmentation of the respective tissues, they were preincubated (30 min) and then incubated (60 min) and total protein was quantified [12]. The glucose uptake results are expressed as nmol glucose/mg of protein. 14C-DG was measured in a liquid scintillator spectrometer (model LS 6500; Multi-Purpose Scintillation Counter-Beckman, Boston, MA, USA).

### 2.5. Action of Cinnamaldehyde on ^14^C-Glucose Uptake in Insulin-Resistant Rats

Slices of soleus muscle from normoglycaemic rats were allocated to the control and treated groups (four slices for each group). The muscles were dissected, preincubated (30 min) and then incubated (60 min) at 37 °C in O_2_/CO_2_ (95%: 5%, *v*/*v*), with pH 7.4 in Krebs Ringer-bicarbonate buffer (KRb) without (control) or with CIN (100 μM CIN). CIN was added to the incubation medium (60 min) in the presence or absence of 300 nM of capsaiscin, 100 µM of alil-isotiocianato (AITC), 100 nM of wortmannin, 10 µM of SB239063, 1 µM of Nifedipine, 50 µM of BAPTA-AM, 1 µM of colchicine, 10 µM of chytocalasin and 50 µM of RO-310432. 14C-DG (0.1 μCi/mL; 0.12 nM) was added to all samples throughout the incubation period, The glucose uptake results are expressed as nmol glucose/mg of protein. 14C-DG was measured in a liquid scintillator spectrometer (model LS 6500; Multi-Purpose Scintillation Counter-Beckman, Boston, MA, USA).

### 2.6. Role of Cinnamaldehyde on ^45^Ca^2+^ Influx

Slices of soleus muscle were pre-incubated for 30 min in an incubator in KRb-HEPES buffer with 5.6 mM of glucose and 0.1 μCi/mL ^45^Ca^2+^ concentrations. [8]. Subsequently, muscles were incubated in the absence (control) or presence of CIN in KRb-HEPES. In some assays, an intracellular calcium chelator or a blocker of L-type voltage-dependent calcium channels was added during the last 60 min of incubation, prior to treatment, and maintained throughout the incubation period. Drugs used included 50 µM of BAPTA-AM and 1 μM of nifedipine and these were processed according the work of Batra et al. [13].

### 2.7. Effect of Cinnamaldehyde on Serum Lipid Profile of Insulin-Resistant Rats

To evaluate the effect of CIN on lipid profile, blood samples from rats induced to insulin resistance by dexamethasone, from control and treated rats, were collected after euthanasia, and the concentrations of triacylglycerol, total cholesterol, low density lipoprotein and high density lipoprotein were determined using kits (Biotécnica, Varginha, Brazil) and expressed as mg/dL [14].

### 2.8. Role of Cinnamaldehyde on the Oral Triglyceride Tolerance Test

Fasted rats (12 h) were separated into groups of six: Group 1 received 6 mL of saline (v.o.); Group 2 received the emulsion (6 mL of corn oil, 80 mg of cholic acid, 2 g of chloestryloleate plus 6 mL of saline) and Group 3 received 20 mg/kg CIN (i.p.) plus lipid emulsion 30 min later. Blood was analyzed at zero time (before the administration of saline, emulsion or CIN) and at 1, 2, 3, 4, 5 and 6 h after emulsion overload. The triglycerides were determined in the blood by a kit from Gold analyses (Belo Horizonte, MG, Brazil).

### 2.9. Statistical Analysis

Data are expressed as means ± standard errors of the mean. One-way analysis of variance and Bonferroni post hoc test or unpaired Student’s *t*-test were used. Differences of *p* ≤ 0.05 were considered as significant using GraphPad Prism^@^ 5 program.

## 3. Results

### 3.1. Cinnamaldehyde and Insulin Tolerance

The synthetic glucocorticoid, dexamethasone, is widely used as a tool to induce insulin-resistance in experimental model rats [9,11]. The present study shows that dexamethasone (0.1 mg/kg; subcutaneously injected for 5 days) was able to induce insulin resistance in rats, compared with the control. Furthermore, the treatment of insulin-resistant rats with CIN (20 mg/kg by intraperitoneal injection, for 5 days) significantly improved insulin-resistance, as shown in Figure 1.

### 3.2. Cinnamaldehyde on Hepatic Glycogen Accumulation

CIN, an essential oil extracted from cinnamon, exhibits several biological effects. Among these effects, CIN achieves carbohydrate balance by inducing insulin and glucagon-like peptide-1 secretion, the inhibition of intestinal disaccharidases, and, consequently, improves glucose tolerance. Figure 2 shows that following in vivo treatment with CIN (20 mg/kg), hepatic glycogen levels in insulin-resistant rats were restored to normal, as compared with control groups.

### 3.3. Effect of Cinnamaldehyde on ^14^C-Glucose Uptake

As previously demonstrated, CIN is an insulin and GLP-1 secretagogue. As such, we investigated the action of CIN on glucose uptake in the soleus and adipose tissue of insulin-resistant rats. Figure 3A,B show that, in both insulin-target tissues, CIN reversed the effect of insulin resistance on glucose uptake.

### 3.4. Mechanism of Action of Cinnamaldehyde on ^14^C-Glucose Uptake

One desirable effect sought in compounds in development for diabetes therapy is their insulin-mimetic action, to rapidly improve glycemic control. Based on our previous studies of calcium influx in β-cells, the concentration of 100 µM of CIN was assayed to investigate the mechanism by which CIN affects glucose uptake in the soleus muscle. Figure 4A shows that CIN (acute effect) has a stimulatory effect on glucose uptake in insulin-resistant. In addition, capsaicin, an agonist of TRPV1, as well as the combination of 300 nM of capsaicin plus CIN, similarly increased glucose uptake to the level observed for the CIN group. Furthermore, a non-electrophilic agonist of TRPA1 (100 µM of AITC) increased glucose uptake, similar to that observed when muscle was incubated with both (CIN + AITC). Figure 4B shows that the action of CIN was significantly reduced (by approximately 40%) in the presence of 100 nM of wortmannin, an inhibitor of PI3-K, and when co-incubated with 10 µM of SB 239063, an inhibitor of p38 MAPK. To investigate the influence of calcium in the mechanism of CIN on glucose uptake, an inhibitor of voltage-dependent calcium channels, 1 µM of nifedine, and an intracellular calcium chelating agent, 50 µM of BAPTA-AM, were added into the medium. Figure 4C shows that the stimulatory effect of CIN was reduced by 45%.

The integrity of the cell cytoskeleton was investigated using an inhibitor of microtubules polymerization, colchicine. When the polymerization of microtubules was interrupted by colchicine, the action of CIN on glucose uptake was blocked around 37%. However, 10 µM of cytochalasin D, an inhibitor of actin association/dissociation, did not alter CIN-induced glucose uptake (Figure 4D). Finally, the influence of protein kinase C (PKC) in the effect of CIN on glucose uptake was investigated using a PKC inhibitor, 50 µM of RO-310432. This drug reduced the effect of CIN on glucose uptake by 30% (Figure 4E).

### 3.5. Effect of Cinnamaldehyde on ^45^Ca^2+^ Influx in Skeletal Muscle

As calcium is mandatory for triggering the trafficking of vesicles for exocytosis, among other mechanisms [15], the role that calcium plays in the stimulatory effect of CIN on glucose uptake was evaluated. Figure 5 shows that both calcium from intracellular stores and extracellular calcium participate in the intracellular signaling of CIN on skeletal muscle, since the presence of BAPTA-AM and nifedipine significantly decreased the CIN-induced calcium influx.

### 3.6. In Vivo Effect of Cinnamaldehyde on Serum Lipid Profile

As expected, the lipid profile behaved in line with the effects of dexamethasone (0.1 mg/kg) after 5 days of subcutaneous injection. Surprisingly, CIN significantly diminished triglycerides, cholesterol and LDL, while improving HDL, thereby ameliorating the total serum lipid profile in insulin-resistant rats after in vivo treatment (Figure 6).

### 3.7. Effect of Cinnamaldehyde on Oral Triglyceride Tolerance Test

The acute effect of CIN on triglyceride tolerance was evaluated from 1 h to 6 h in normal rats overloaded with a fat-enriched emulsion, as shown in Figure 7. The progressive increase in serum triglycerides observed in the lipid-enriched emulsion-overloaded group was significantly reduced by CIN (20 mg/kg i.p.) from 2 h to 6 h (Figure 7).

## 4. Discussion

An insulin receptor can detect high glucose and activates intracellular downstream signaling pathways to regulate glycemia [16]. This augments glucose uptake, particularly in insulin-target tissues, through the action of increased glucose transporter-4 (GLUT-4). The pleiotropic insulin receptor depends on intracellular substrates to trigger specific mechanisms to increase GLUT-4 at the plasma membrane, such as that mediated by PI3-K. In the present study, CIN, an electrophilic agonist of TRPA1, improved insulin tolerance after 5 days of in vivo treatment. This is in line with reports in the literature that suggest several therapeutic effects of TRPA1, including increased glucose uptake, improved glucose insulin sensitivity and heart protection [17]. Furthermore, the effect of CIN on insulin sensitivity tissues may contribute to its ability to improve insulin-resistance, as reported by Zhu et al. [18]. Our findings further support the ability of CIN to improve insulin resistance, as shown by analyzing hepatic glycogen accumulation and glucose uptake in muscle and adipose tissue. In diabetic rats, CIN has been reported to increase glucose uptake and improve insulin sensitivity in adipose and muscle, increase glycogen synthesis in the liver, restore pancreatic cell function, delay gastric emptying and improve diabetic renal and brain disorders through multiple methods [17,18].

We also investigated how CIN modulated glucose uptake in insulin-resistant muscle using ^14^C-glucose. Results indicate that CIN may increase glucose uptake via a TRPA1-mediated mechanism and through TRPV1. In addition, the participation of PI3-K, p38 MAPK and PKC is crucial for the complete stimulatory effect of this compound on glucose uptake in skeletal muscle. Zhu et al. [18] reported that PI3-K and p38 MAPK participate in the CIN-induced signaling pathway that increases glucose uptake in diabetic rats, both in muscle and in adipose tissue, through the TRPA1/ghrelin receptor. Among the various phosphorylation events that are mediated by PKC, the activity ionic channels are regulated by a family of kinases, including the T-type VDCC, and the transient receptor potential melastatin family (TRPM) [19].

We also studied the ionic influence on the stimulatory effect of CIN on glucose uptake. Incubation of tissues with nifedipine, and with BAPTA-AM, demonstrated a primary signaling role for calcium (extracellular and intracellularly stored calcium) in CIN-induced glucose uptake. A major PKC family is calcium dependent or calcium-calmodulin dependent, coordinating the activity of ionic channels to trigger several physiological events [19,20,21].

The cytoskeleton supports and organizes the trafficking of organelles and vesicles. The cytoskeleton and actin movement are essential for the intracellular displacement of vesicles and the distribution or secretion of proteins intra- and inter-cellularly [22,23]. When CIN was co-incubated with colchicine, a microtubule disruptor, the stimulatory effect of the compound was significantly diminished. However, under similar experimental conditions, cytochalasin D, an inhibitor of actin polymerization, did not affect glucose uptake. As currently discussed in the literature, the functional integrity (polymerization/depolymerization cycle) of microtubules and actin is essential to vesicle trafficking. Mendes et al. [14] showed that the integrity of the microtubule network is necessary for the translocation of insulin vesicles from the cytosol to the plasma membrane in pancreatic islets. Similarly, in other systems, such as the central nervous system, the integrity of the cytoskeleton (microtubule or actin network) is essential for the recycling of synaptic vesicles [24].

Insulin resistance is associated with a metabolic condition that leads to lipid and carbohydrate homeostasis impairment. Significant diminution in triglycerides and LDL was detected in insulin-resistant rats treated with CIN for 5 days. This occurred in association with an augment in HDL, demonstrating effects of CIN on lipid homeostasis. Additionally, acute CIN exposure significantly improved triglyceride tolerance, with the effect lasting for 2 h to 6 h after lipid overloading. Our findings of improvements in lipids after CIN treatment support the data discussed by Jamali et al. [25], Allen et al. [26] in studies from type 2 diabetic patients, and Li et al. [27] in db/db mice. However, these data are conflicting and require further mechanistic investigation.

## 5. Conclusions

The effect of cinnamaldehyde on insulin resistance is mediated by TRPA1, with downstream signaling involving the activation of PI3-K, MAPK, PKC, as well as extracellular and stored calcium. Complete cytoskeleton integrity is required for its full stimulatory effect on glucose uptake in soleus muscle. Additionally, cinnamaldehyde ameliorates the lipid profile and improves triglyceride tolerance following acute in vivo exposure.

## Figures and Tables

**Figure 1 nutrients-17-00297-f001:**
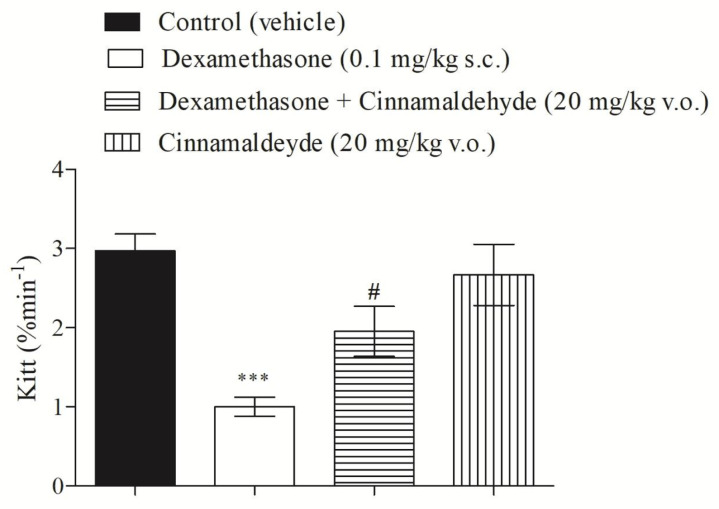
Effect of CIN (5 days, 20 mg/kg) on the insulin-tolerance test in insulin-resistant rats. N = 5, *** *p* ≤ 0.001 compared to control (vehicle); ^#^ *p* ≤ 0.05 compared to dexamethasone group.

**Figure 2 nutrients-17-00297-f002:**
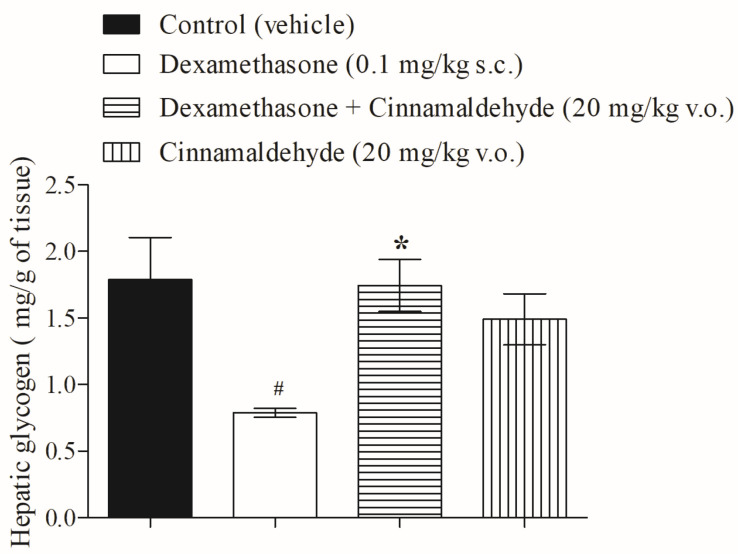
Action of CIN on hepatic glycogen accumulation in insulin-resistant rats, after 5 days of in vivo treatment. N = 5, ^#^ *p* ≤ 0.05 compared with control (vehicle); * *p* ≤ 0.05 compared with the dexamethasone group.

**Figure 3 nutrients-17-00297-f003:**
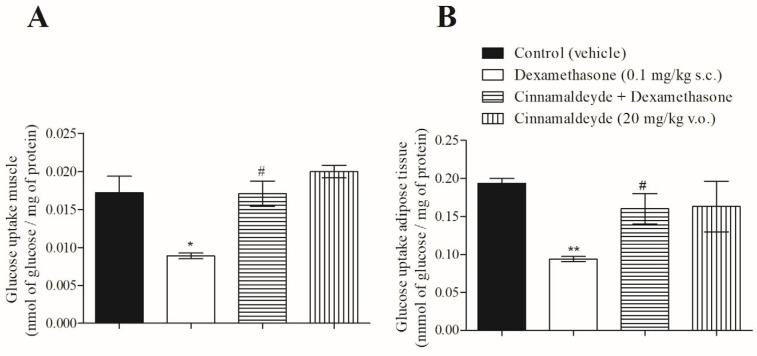
Effect of CIN on [^14^C]-glucose uptake in (**A**) soleus skeletal muscle and (**B**) adipose tissue from insulin-resistant rats, treated for 5 days. N = 5, * *p* ≤ 0.05 and ** *p* ≤ 0.01 compared with control (vehicle); ^#^ *p* ≤ 0.05 compared with dexamethasone group.

**Figure 4 nutrients-17-00297-f004:**
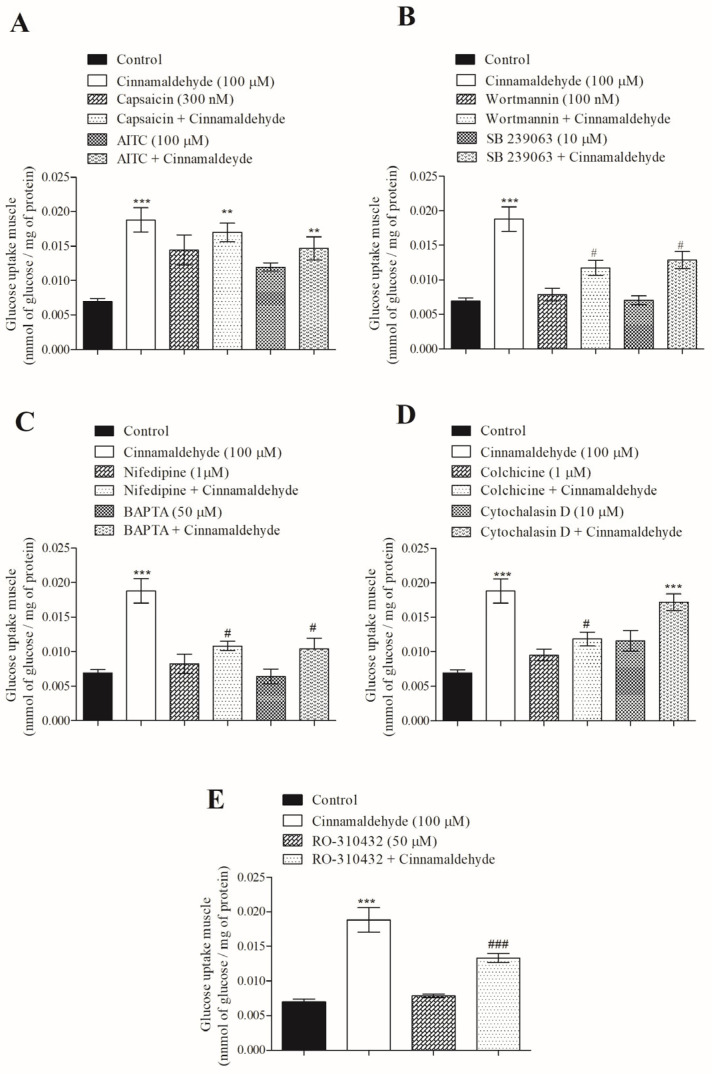
Mechanism of action by which CIN induces [^14^C]-glucose uptake in skeletal muscle from insulin-resistant rats. Involvement of TRPV1 (capsaicin) and TRPA1 (AITC) (**A**); PI3-K (wortmannin) and p38 MAPK (SB 239063) (**B**); L-type VDCC (nifedipine) and calcium from stores (BAPTA-AM) (**C**); cytoskeleton (colchicine) and actin (cytochalasin) (**D**); and PKC (RO-3104320) (**E**). Pre-incubation time = 30 min; incubation time = 60 min. N = 6 for each group. *** *p* ≤ 0.001 and ** *p* ≤ 0.01 compared to control group. ^###^ *p* ≤ 0.001 and ^#^ *p* ≤ 0.05 compared to CIN group.

**Figure 5 nutrients-17-00297-f005:**
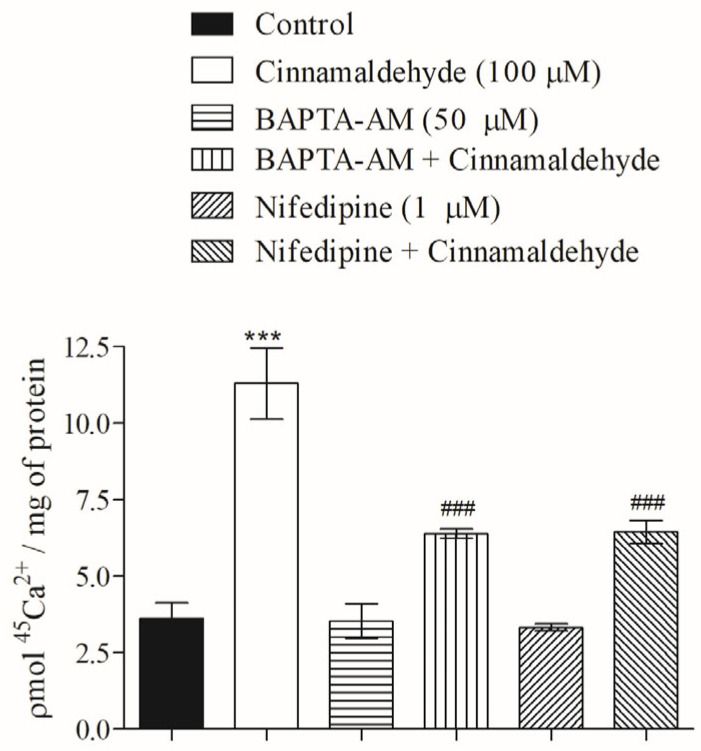
Mechanism of action of CIN on ^45^Ca^2+^ influx in skeletal muscle. Involvement of calcium from intracellular stores (BAPTA-AM) and extracellular calcium by L-type VDCC (nifedipine). Pre-incubation time = 30 min; incubation time = 60 min. N = 6 for each group. *** *p* ≤ 0.001, compared to control group and ^###^ *p* ≤ 0.001, compared to CIN group.

**Figure 6 nutrients-17-00297-f006:**
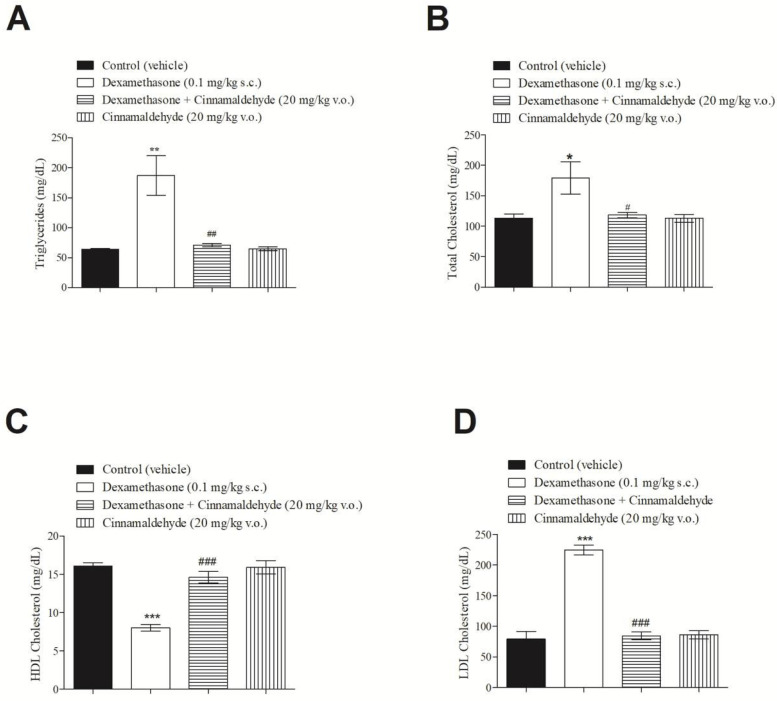
Effect of CIN on (**A**) triglycerides, (**B**) total cholesterol, (**C**) HDL cholesterol and (**D**) LDL cholesterol in insulin-resistant rats after 5 days of treatment with dexamethasone. N = 6 for each group. *** *p* ≤ 0.001; ** *p* ≤ 0.01 and * *p* ≤ 0.05, compared to the control group. ^###^ *p* ≤ 0.001; ^##^ *p* ≤ 0.01 and ^#^ *p* ≤ 0.05, compared to the dexamethasone group.

**Figure 7 nutrients-17-00297-f007:**
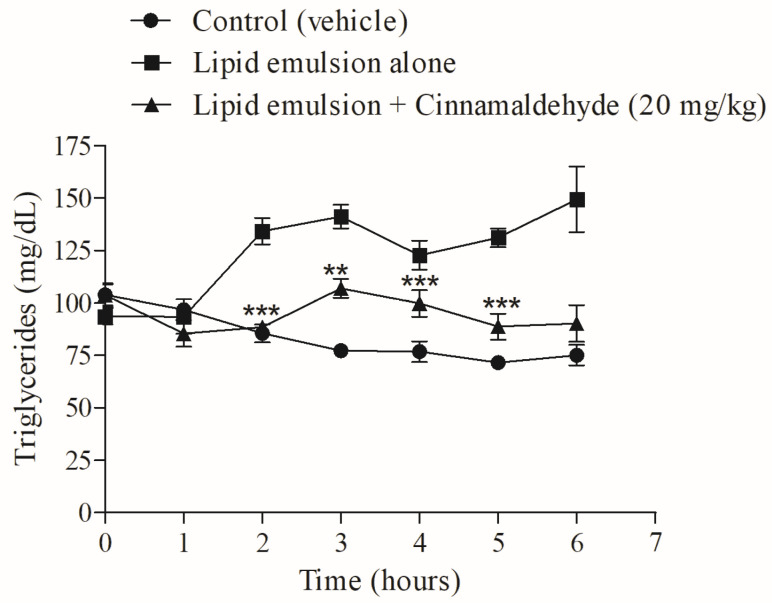
Acute effect of CIN on the oral triglyceride tolerance test in normal rats overloaded with an enriched lipid emulsion, at 0 to 6 h. N = 6. *** *p* ≤ 0.001 and ** *p* ≤ 0.01, compared to the respective value for the lipid emulsion group.

## Data Availability

All data in this study are contained within the article.

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
