# Peer review of "Potential Effect of Cinnamaldehyde on Insulin Resistance Is Mediated by Glucose and Lipid Homeostasis"

_nutrients, 2025, doi:10.3390/nu17020297_

Round 1

Reviewer 1 Report

Comments and Suggestions for Authors

I consider that this paper requires major revision before being accepted. Some clarifications should previously been clarified in relation to this study.

Major comments

- Inclusion and exclusion criteria should be described in more detail.

- Authors refer to 5 or 6 rats in the different experimental groups. Could authors explain the reason for this difference? Additionally, authors should explain if randomization has been performed, because for this reviewer it is not clear.

- Experimental animals. The provenance of animals should be explained in more detail (city, country). Authors should justify why they only used male rats.

- Why did authors only use 5/6 animals for each experimental group?

- Statistical methods. Authors should describe if obtained data get the parametric or non-parametric conditions and therefore correlate them with the specific tests used for the analysis of variance. Authors refer that data were expressed as mean ± SEM; however after they report that data are sampled from all population with the same standard deviation (SD). Authors should explain the reason for using both, I mean SEM and SD. Therefore, more detailed information of the GraphPad Prism@ 5 program used in the statistical analysis should be supplied.

- Why, to perform tests with the insulin-resistance model, rats were treated only for 5 consecutive days? Do authors think that similar results would be obtained after one/two months of treatment?

- The limitations of the study should be included in the revised version.

- In the conclusion’ section, authors should clearly explain which are the main findings of this research in comparison with those of others previously performed.

- Authors should carefully read the instructions for authors of Nutrients specifically the section referred to the way to cite the references in order to unify all them. Sometimes they use the whole name of the Journal, others the simplified one, the change the location of the year of publication…

Minor comments

- Line 86. Authors wrote “(20/mg/kg)”. Do they mean to “(20 mg/kg)”? This amount was daily administered or corresponds to the whole 5 days during which the study was performed? Please clarify.

- Line 99, 109, 123. Authors should write “O2/CO2” with subscripts.

- Line 123. Authors should write “45Ca2+” with properly with superscripts and subscripts.

-Line 187, 203... Authors write “Figure” and “Fig.”, respectively. Please, unify along the ms.

- Line 261. Authors wrote “(Fig.7).” Please correct and leave one space where corresponds.

Author Response

The references self-citations were now according to the journal requirement (about 14.81%). The marked version (word) shows all revision made and is highlighted in blue.

Reviewer 2 Report

Comments and Suggestions for Authors

The authors report that the potential effects of cinnamaldehyde on insulin resistance are mediated by glucose and lipid homeostasis. The potential effects of cinnamaldehyde on insulin resistance are mediated by glucose and lipid homeostasis.

-Specific clarification is needed on whether cinnamaldehyde is an agonist or antagonist to the TRPA1 receptor in type 1 diabetes.

- Whether cinnamaldehyde can modulate beta cells and, if so, whether it can treat type 1 diabetes

-The TRPA1 receptor has many different agonists, is it the same signaling pathway as cinnamaldehyde.

-What anesthetics were used and how does your institution manage animals postmortem from an animal ethics perspective?

Author Response

(The authors gave the same response as above.)

Round 2

Reviewer 1 Report

Comments and Suggestions for Authors

Accept.

Reviewer 2 Report

Comments and Suggestions for Authors

Well done!